The impact of the filling technique with two sealers in bulk or associated with gutta-percha on the fatigue behavior and failure patterns of endodontically treated teeth

Lena Isabella Marian 1
Chiaratti Luiza Colpo 2
http://orcid.org/0000-0002-7853-5967 Pilecco Rafaela Oliveira 3
Machry Renan Vaz 4
http://orcid.org/0000-0002-5412-3546 Tribst João Paulo Mendes 5 j.p.mendes.tribst@acta.nl
http://orcid.org/0000-0002-3218-8031 Kleverlaan Cornelis Johannes 6
http://orcid.org/0000-0002-9077-9067 Pereira Gabriel Kalil Rocha 1
Morgental Renata Dornelles 1
1 Post-Graduate Program in Oral Science, Universidade Federal de Santa Maria , Santa Maria, Rio Grande do Sul , Brazil
2 Faculty of Dentistry, Universidade Federal de Santa Maria , Santa Maria, Rio Grande do Sul , Brazil
3 Department of Conservative Dentistry, Faculty of Dentistry, Federal University of Rio Grande do Sul , Porto Alegre, Rio Grande do Sul , Brazil
4 Department of Restorative Dentistry, Federal University of Minas Gerais , Belo Horizonte, Minas Gerais , Brazil
5 Department of Reconstructive Oral Care, University of Amsterdam , Amsterdam, Noord Holland , Netherlands
6 Department of Dental Materials Science, University of Amsterdam , Amsterdam, Noord Holland , Netherlands
Abu Hasna Amjad
Electronic publication date: 2024 Oct 28
Publication date: 2024
Volume: 12
Electronic Location ID: e18221
Received 2024 Jun 20; Accepted 2024 Sep 11
Copyright: © 2024 Lena et al.
Copyright year: 2024
Copyright holder: Lena et al.
License: This is an open access article distributed under the terms of the Creative Commons Attribution License, which permits unrestricted use, distribution, reproduction and adaptation in any medium and for any purpose provided that it is properly attributed. For attribution, the original author(s), title, publication source (PeerJ) and either DOI or URL of the article must be cited.
License URL: https://creativecommons.org/licenses/by/4.0/

Keywords: Fatigue, Endodontic sealers, Endodontically treated teeth, Root canal filling, Root fracture

Funding: CAPES Finance code 001 Brazilian National Council for Scientific and Technological Development–CNPq #140118/2022-5; Post-Doctoral Scholarship #151004/2022-6 Research Productivity Scholarship Pq2 #304665/2022-3 This study was financed by the Brazilian Federal Agency for Coordination of Improvement of Higher Education Personnel–CAPES (Isabella Marian Lena Doctorate’s scholarships; Finance code 001), and by the Brazilian National Council for Scientific and Technological Development–CNPq (Rafaela Oliveira Pilecco’s doctorate scholarship, project #140118/2022-5; Renan Vaz Machry’s post-doctoral scholarship #151004/2022-6; Gabriel Kalil Rocha Pereira’s research productivity scholarship Pq2, process number #304665/2022-3). There was no additional external funding received for this study. The funders had no role in study design, data collection and analysis, decision to publish, or preparation of the manuscript.

==============================
Background

The present in vitro study aimed to evaluate the fatigue behavior of teeth filled with a calcium silicate-based sealer (Bio-C Sealer, BC) or an epoxy resin-based sealer (AH Plus, AH), in bulk or associated with gutta-percha as main core material.

Methods

Seventy-two sound human maxillary anterior teeth were initially selected. Sixty of them, were randomly chosen, and had their root canals prepared using nickel-titanium reciprocating instruments, being again randomly assigned to five experimental groups (n = 12): C+ (control + prepared but not filled); BC-B (BC in bulk); BC-GP (BC+ gutta-percha); AP-B (AH in bulk); AP-GP (AH+ gutta-percha). An additional negative control group (C−) was considered (n = 12), consisting only on sound teeth, without preparation and filling. The specimens were submitted to a survival analysis after the cyclic fatigue test.

Results

Sound teeth (C−) presented the best fatigue performance (P < 0.05), being similar only to the AP-GP group (P > 0.05). Despite that, all experimental groups showed similar fatigue behavior (P > 0.05) to C+ (BC-B = BC-GP = AP-B = AP-GP = C+). Based on that, it can be seen that the use of gutta-percha, as a main core material, associated with the AH Plus sealer, reestablished the mechanical fatigue performance of endodontically treated teeth comparable to sound teeth, still consisting on the most promising approach to rehabilitate such scenario. Teeth filled in bulk, had discreetly higher risk of premature failures and inferior fatigue performance.

Introduction

Root fracture (RF) is one of the possible consequences of endodontic treatment, representing a challenging diagnostic situation that often leads to tooth extraction (Patil et al., 2017). Potential causes of RF include excessive loss of tooth structure due to caries, trauma and endodontic procedures, such as root canal access and instrumentation, or exaggerated stress induced during lateral condensation of gutta-percha (Patel, Bhuva & Bose, 2022). Furthermore, the microstructural alterations in the dentin can adversely affect the fatigue behavior of endodontically treated teeth by the growth and coalescence of intrinsic defects, a critical concern given the cyclic loads these teeth endure (Barreto et al., 2012; Mannocci et al., 2022). In this sense, one of the main goals of endodontic filling should be to seal the root canal with materials that enhance the fatigue strength of the remaining tooth structure (Osiri et al., 2018) ultimately prolonging the life of the tooth.

The standard obturation technique involves using an endodontic sealer associated with a main core material. Gutta-percha is a thermoplastic material widely used to fill root canals due to its good biological behavior and low allergenic potential; it is also easily removed in cases of endodontic reintervention (İnce Yusufoglu et al., 2019). However, due to its low elastic modulus and inability to chemically bond to dentin walls, the use of endodontic sealers alongside gutta-percha is essential (Hammad, Qualtrough & Silikas, 2009). The sealers promote the union between the main core material (gutta-percha) and the root canal walls. Furthermore, they have the ability to penetrate into dentinal tubules, establishing a micromechanical interlocking that might increase the fatigue resistance of the roots (Osiri et al., 2018). AH Plus (Dentsply DeTrey GmbH, Konstanz, Germany), an epoxy-resin sealer, is regarded as the ‘gold-standard’ for root canal filling. This is attributed to its outstanding physicochemical properties (Zhou et al., 2013) and satisfactory biological performance (Cintra et al., 2017). Studies have demonstrated that AH Plus increases the fracture resistance of root-canal treated teeth when compared to other materials (Topçuoğlu et al., 2013; Osiri et al., 2018).

Currently, tricalcium silicate-based sealers are available in the market, recognized due to their biocompatibility, antimicrobial and bioactive properties (Sfeir et al., 2021). Some authors indicate that calcium silicate sealers could be successfully used alone in the root canal space as a single root filling material (i.e., in bulk) or associated with a main core material like gutta-percha (Nagas et al., 2014; Eymirli et al., 2019). Nagas et al. (2014) state that in the first scenario, the sealer adheres more uniformly to dentin, potentially forming a ‘monoblock’ along the root canal and thereby increasing the dislocation resistance of the filling (Nagas et al., 2014). Bio-C Sealer (Angelus, Londrina, Paraná, Brazil) is a Brazilian calcium silicate-based sealer provided by the manufacturer in an injectable and pre-mixed syringe that requires moisture from the dentinal tubules to promote the total setting (López-García et al., 2019). Bio-C Sealer has been extensively studied in the literature and is noted for its ability to alkalize the environment, short setting time, suitable radiopacity, and minimal volumetric change (Tolosa-Monfà et al., 2023; Kwak et al., 2023). Compared to AH Plus (Dentsply), Bio-C offers a superior flow rate which enhances its capacity to fill spaces and irregularities within the root canal system (Girelli et al., 2023). Nonetheless there is still a need for more scientific research on the fatigue behavior of teeth filled with this sealer material.

Furthermore, few studies have explored root canal obturation using endodontic sealers in bulk, without a main core material (Jainaen, Palamara & Messer, 2007; Nagas et al., 2014). Thus, the aim of the present in vitro study was to evaluate the fatigue behavior of root-canal treated teeth filled by the single cone technique (with gutta-percha) or by the “in bulk” technique (without gutta-percha) with the new endodontic sealer (Bio-C Sealer), compared to the “gold standard” (AH Plus). The null hypothesis is that there will be no difference between the fatigue resistance values with the different sealers and filling techniques.

Materials and Methods

Study design

The manuscript was written based on the ‘Preferred Reporting Items for Laboratory studies in Endodontology (PRILE) 2021’ guideline (Nagendrababu et al., 2021). The study was submitted and approved by the Institutional Ethics Committee (CAAE 30345120.7.0000.5346) from the Research Ethics Committee at the Federal University of Santa Maria.

The sample size calculation was performed based on the parameters described by Patil et al. (2017) (monotonic test): fatigue failure load of 332 (±17) N for group 1 (AH Plus) and 354 (±20) N for group 2 (Bio-C Sealer); 80% power; 5% significance level (OpenEpi 3.01, Atlanta, GA, USA). Therefore, the estimated minimum sample size was found to be 12 teeth per group (n = 12).

This study was conducted with two control groups established based on previous study designs (Topçuoğlu et al., 2013; Patil et al., 2017), and four experimental groups, considering the “endodontic sealer” factor in two levels (AH-Plus and Bio-C Sealer) and the “filling technique” factor in two levels (with or without gutta-percha). Thus, teeth were randomly divided into six groups as follows (N = 72): C−: no intervention. C+: root canal preparation but no filling; BC-B: root canal filling using Bio-C Sealer in bulk; BC-GP: root canal filling using Bio-C Sealer and the single cone technique (Reciproc R50 gutta-percha point; VDW); AP-B: Root canal filling using AH Plus sealer in bulk; AP-GP: Root canal filling using AH Plus sealer and the single cone technique (Reciproc R50 gutta-percha point; VDW).

Specimen preparation

Seventy-two sound human maxillary anterior teeth with complete root formation and straight roots (<5°) were used. To confirm the presence of a single root canal, the teeth were submitted to digital periapical radiographs (RVG 5100; Carestream Health, Rochester, NY, USA) in the buccolingual (BL) and mesiodistal (MD) directions. With the aid of a digital measurement tool (ImageJ; U.S. National Institutes of Health, Bethesda, MD, USA), the BL and MD dimensions were obtained and the average values were calculated. Teeth that showed measurements 15% greater than the average were discarded to obtain a more homogeneous sample (Osiri et al., 2018). The external root surfaces were cleaned with periodontal scalers (Golgran, São Paulo, SP, Brazil) and analyzed under a digital stereomicroscope (StereoDiscovery V20; Zeiss, Oberkochen, Germany) with 20× magnification to verify the presence of pre-existing defects. In case of cracks or fracture lines, the teeth were excluded and replaced.

Twelve sound teeth were than randomly allocated to the C− group, and the sixty remaining had their root canals accessed and prepared by a single trained operator (L.C.C). After endodontic access, the working length (WL) was determined by subtracting 1 mm from the length of a #10 K-file (Dentsply Maillefer, Ballaigues, Switzerland) with its tip visualized at the apical foramen. Root canal preparation was performed using nickel-titanium reciprocating instruments (Reciproc R50; VDW, Munich, Germany) in the mode ‘‘Reciproc All’’ on an endodontic motor (VDW Silver; VDW, Munich, Germany). In-and-out movements with an amplitude of 3 mm were applied in the cervical, middle and apical root thirds until reaching the WL. Throughout instrumentation, the root canals were irrigated with 20 ml of 2.5% sodium hypochlorite solution (NaOCl; Asfer Indústria Química, São Caetano do Sul, SP, Brazil), followed by 2 ml of 17% EDTA (Biodinâmica, Ibiporã, PR, Brazil) for smear layer removal. Then, 5 ml of saline solution (Farmax, Divinópolis, MG, Brazil) were used as final irrigation.

At this moment, the sixty teeth were randomly allocated into the remaining five experimental groups (C+, BC-B, BC-GP, AP-B, and AP-GP). After, all teeth specimens were embedded in polyvinyl chloride (PVC) cylinders (Dencrilay; Dencril, Caieiras, SP, Brazil) (20 mm × 25 mm) filled with a chemically cured acrylic resin (Clássico, Campo Bom Paulista, SP, Brazil), as previously described by Osiri et al. (2018). The specimens were fixed in a parallelometer, with the long root axis of the teeth and cylinder parallel to each other and perpendicular to the ground. Then, the acrylic resin was poured inside the cylinder up to 3 mm from the cement-enamel junction. For the groups filled with AH Plus, the root canals were totally dried with #50 absorbent paper points (Tanari, Manacapuru, AM, Brazil). On the other hand, for the groups filled with Bio-C Sealer, the canals were dried with only one absorbent article point to maintain residual moisture (Nagas et al., 2012; Pelozo et al., 2023). Endodontic sealers and their compositions are described in Table 1.

Table 1 Tested materials and their composition.

Material	Composition	Manufacturer	
AH Plus Jet	Paste A: bisphenol A epoxy resin; bisphenol-F epoxy resin; calcium tungstate; zirconium oxide; iron oxide and silica.
Paste B: adamantized amine; N, N″-dibenzyl-5-oxanonanediamine 1,9; TCD-diamine; calcium tungstate; zirconium oxide; silicone oil and silica.	Dentsply; DeTrey GmbH, Konstanz, Germany.	
Bio-C Sealer	Tricalcium silicate; dicalcium silicate; tricalcium aluminate; calcium oxide; zirconium oxide; silicon oxide; polyethylene glycol; iron oxide.	Angelus; Londrina, Paraná, Brazil.	

In the BC-B and BC-GP groups, Bio-C Sealer was introduced into the root via an intracanal tip provided by the manufacturer. Then, a #20 K-file was used to spread the sealer on the canal walls with pumping movements, followed by a Lentulo spiral (N° 3; Dentsply-Maillefer, Ballaigues, Switzerland) at low speed, until the root was completely filled (Pinto et al., 2021). Additionally, only in the BC-GP group, a Reciproc R50 gutta percha point (VDW), was selected, and their radiographic fit was verified to ensure adaptation 1 mm short of the apex. The cone was then embedded with the sealer and slowly inserted into the canal until it reached the WL.

In the AP-B and AP-GP groups, AH Plus was dispensed directly into the root canal from the double-barrel mixing syringe via an intracanal tip attached to the auto-mixing tip provided by the manufacturer. Next, the sealer was spread on the canal walls with the aid of an #20 K-file and a Lentulo spiral, as previously described. Additionally, only in the AP-GP group, a Reciproc R50 gutta percha point (VDW) was selected, and their radiographic fit was verified to ensure adaptation 1 mm short of the apex. The cone was then embedded with the sealer and slowly inserted into the canal until it reached the WL.

Mesiodistal and buccolingual digital periapical radiographs were taken to ensure no empty spaces in the obturation mass. After the complete setting of the sealer, the specimens were restored. First, the endodontic access was etched with 37% phosphoric acid for 15 s, then rinsed and gently air-dried, followed by the application of a bond agent (Single Bond; 3M ESPE, St Paul, MN, USA). Then, a resin composite (Filtek Z350 XT; 3M, Sumaré, São Paulo, Brazil) was inserted by the incremental technique and light-cured (1,200 mW/cm2, Radii Plus; SDI, Bayswater, Australia) for 30 s on the palatal side (Missau et al., 2017).

Cyclic fatigue testing

All specimens were submitted to the cyclic fatigue testing in an electro-dynamic testing machine (Instron ElectroPlus E3000; Instron Corporation, Norwood, MA). A 45° angle load was applied by a 6-mm-diameter stainless-steel sphere, directly positioned into the lingual/palatal surface while the specimens were submerged in distilled water, as previously described by Missau et al. (2017). The fatigue test was performed with an initial load of 100 N at a frequency of 20 Hz for 5,000 cycles (to ensure the correct positioning of the piston over the specimen), followed by increments of 25 N for 10,000 cycles in each step, until the specimen failure. At each specimen step the specimen was transilluminated to search for initial cracks, that could be considered as failure, but as they were not seen, the test stopped only with the complete fracture of the teeth. The fatigue failure load (FFL) and the number of cycles for failure (CFF) were recorded for statistical survival analysis.

Failure mode

For fracture morphology analysis, the specimens were analyzed under a stereomicroscope (Stereo Discovery V20; Zeiss, Gottingen, Germany) under 4× magnification by a trained and calibrated examiner (L.C.C). The training consisted of an expository lecture by an experienced examiner (I.M.L). Calibration was performed using 20% of the study sample. Inter- and intra-examiner agreement was calculated using Cohen’s kappa coefficient (>0.76).

The failure mode was classified according to the following criteria adapted from Missau et al. (2017) (26): mode I, fracture above the cement-enamel junction (repairable); or mode II, fracture below the cement-enamel junction (irreparable). Representative imagens of the failure modes were obtained using a digital camera (Fig. 1).

Figure 1 Representative images of the failure modes: (A) Mode I, fracture above the cement-enamel junction (reparable); (B) Mode II, fracture below the cement-enamel junction (irreparable).

Data analysis

Data analysis was performed using the SPSS Statistics v. 21 software (IBM Analytics, Chicago, IL, USA), where FFL (in Newtons) and CFF (in counts) measurements were analyzed by Kaplan-Meier and Mantel-Cox (Log Rank) survival tests (P < 0.05) and their survival rates plotted through each step of the test in graphs. Regarding the failure mode data, they were described qualitatively and a chi-square test was applied to evaluate the association between endodontic sealer or filling technique and the type of fracture.

Results

There was a significant difference (P < 0.05) between the studied conditions, where the C- group (sound teeth) showed higher FFL and CFF, which can be translated as a superior fatigue behavior as this group survived more steps before fracture. AP-GP group, was the only experimental group statistically to sound teeth (C- group) (P > 0.05). Despite that, no differences were found between AP-GP and the other experimental groups (Table 2, Figs. 2A and 2B). Such differences are even more clear when considering the survival rates through the fatigue testing (Figs. 2A and 2B), as it was observed that BC-B, AP-B and BC-GP groups presented a higher risk of early failure compared to C+, C− and AP-GP groups since they presented, respectively, 25%, 17% and 17% probability of failure at step 275 N (45,000 cycles), while the latter still held 0% probability of failure (Table 3). Furthermore, it was found that, with the advancement of the fatigue test, at step 475 N (125,000 cycles), all groups that received endodontic preparation showed similar results for probability of failure (ranging from 17% to 33% probability of survival). In contrast, the group that did not receive the intervention (C−) had a survival rate of 100%. In addition, at step 625 N (185,000 cycles), only AP-GP and C- groups presented a considerable survival (33–42%), while the AP-B group already had a 100% probability of failure at step 500 N (135,000 cycles).

Table 2 Mean, standard deviation and 95% confidence interval of fatigue failure load (FFL) (N) and cycles for failure (CFF) in each experimental group.

Groups	FFL (N)	CFF	
Mean ± standard deviation	95% CI	Mean ± standard deviation	95% CI	
BC-B	427.0 ± 139.5B	[338.4–515.7]	100,450 ± 57,389 B	[63,986–136,914]	
BC-GP	414.5 ± 139.9B	[325.6–503.5]	93,795 ± 56,471B	[57,915–129,675]	
AP-B	370.8 ± 87.1B	[315.4–425.2]	77,972 ± 35,910B	[55,156–100,789]	
AP-GP	525.0 ± 231.1A,B	[378.1–671.8]	140,666 ± 94,350A,B	[80,718–200,613]	
C+	450.0 ± 81.1B	[398.4–501.5]	109,953 ± 31,446B	[89,972–129,933]	
C−	687.5 ± 206.5A	[556.2–818.7]	203,177 ± 83,114A	[150,369–255,986]	
Note:

Different uppercase letters in the columns mean statistically significant difference (Kaplan-Meier and Mantel-Cox post-hoc tests) for FFL and CFF (p < 0.05).

Figure 2 Survival plot according to the FFL (A) and CFF (B) for each group.

It is noticeable that sound teeth and teeth sealed with AH-Plus and gutta-percha presented the highest survival rates.

Table 3 Failure mode (n(%)) in each experimental group.

Groups	Reparable	Irreparable	P-value through Chi-square test	
Mode I	Mode II	
BC-B	3 (25)	9 (75)	P > 0.05 indicating no statistical differences among groups	
BC-GP	1 (8.3)	11 (92)	
AP-B	2 (17)	10 (83.3)	
AP-GP	1 (8.3)	11 (91.7)	
C+	0 (0)	12 (100)	
C−	0 (0)	12 (100)	

Mode II failures (irreparable) were the most prevalent in all experimental groups, and no statistical differences were observed among groups using the Chi-square test for failure mode data, indicating that the failure pattern was consistent regardless of the sealer or technique used.

Discussion

Root-canal treated teeth are more susceptible to root fracture than teeth without endodontic treatment (Patel, Bhuva & Bose, 2022). In this context, endodontic sealers are used to promote reinforcement of the remaining structure, adhering to the root canal surface, in an attempt to contribute to the long-term permanence of these teeth (Topçuoğlu et al., 2013; Patil et al., 2017). Thus, the present study evaluated the fatigue behavior of teeth filled with two different endodontic sealers (Bio-C Sealer and AH Plus), with or without a main core material (gutta-percha). Indeed, it was observed that the endodontic access and preparation might induce damage to the root that can only be attenuated by the materials and filling techniques explored. In this sense, the null hypothesis was accepted since no significant difference was found among the experimental groups. However, the AP-GP group was the only one that induced a fatigue behavior similar to sound teeth (C− group), which presented the superior fatigue performance.

In order to maintain a tridimensional seal within the root canal system, a filling material must adhere to the dentin walls (Pirani & Camilleri, 2022). Many studies have shown that epoxy resin-based sealers result in a high bond strength to the root canal dentin (Nagas et al., 2014), penetrating into micro-irregularities and partially filling dentinal tubules (Missau et al., 2017), thus increasing mechanical retention and resistance to shear forces (Sousa-Neto et al., 2005). In the present study, teeth filled with AH Plus (an epoxy resin-based sealer) and gutta-percha (AP-GP group) showed the best fatigue behavior, similar to teeth that did not receive any intervention (group C−). These results were also found in a previous study (Topçuoğlu et al., 2013; Missau et al., 2017), which demonstrated that AH-Plus in association with a gutta-percha point can enhance the fatigue resistance of root-canal treated teeth. Based on that, the filling procedure with AH Plus sealer and the single cone technique improved the teeth’s mechanical performance, resulting in a similar behavior to sound teeth.

Sealers containing calcium silicate represent an important alternative to filling procedures nowadays, especially due to their remarkable biocompatibility. Another feature that made calcium silicate-based sealers so popular is the potential ability to form a chemical bonding to dentin (Sfeir et al., 2021). Although the exact mechanism is still unclear, the nanoparticles present in these materials may allow it to flow into dentinal tubules forming interlocking bonds and establishing a mineral infiltration zone with the posterior formation of hydroxyapatite (Sfeir et al., 2021). Recent studies have demonstrated that Bio-C Sealer showed better cytocompatibility, mineralization capacity (López-García et al., 2019), higher penetration and better adaptation to the dentinal tubules (Caceres et al., 2020) compared to other filling materials. Moreover, it was shown that teeth filled with calcium silicate-based sealers presented higher values for fracture resistance when compared to those filled with AH Plus (Topçuoğlu et al., 2013; Patil et al., 2017). These findings contradict those described in our study, where teeth filled with Bio-C Sealer presented a similar mechanical behavior to AP groups. When compared to sound teeth, filling with Bio-C sealer showed significantly worse fatigue behavior. This may be explained by the different methodologies applied since most studies regarding mechanical behavior of root filled teeth use static load tests, which can be misleading, and we used a fatigue method that, despite its limitations, is the most recommended approach to induce clinically relevant findings (Topçuoğlu et al., 2013; Celikten, Uzuntas & Gulsahi, 2015; Patil et al., 2017).

Tooth fracture is a consequence of cyclic fatigue that happens in the oral cavity in response to the stress caused during mastication. In this situation, failure happens in a much lower load than the maximum load capacity (Ordinola-Zapata & Fok, 2021). The cyclic fatigue test used here simulates, under controlled parameters such as number of cycles, frequency, and load, the intermittent loading movements observed intraorally (Missau et al., 2017). Thus, cyclic fatigue tests provide results replicating the clinical conditions and the cyclic nature of chewing. Studies using static loading only provide the maximum critical stress through the increased applied load, which does not allow predicting failures over time, since they do not simulate the stimuli of the oral cavity (Missau et al., 2017; Ordinola-Zapata & Fok, 2021).

Due to its potential ability to bond to root dentin, the manufacturers claim calcium silicate-based sealers perform successfully regardless of using a main core material (Nagas et al., 2014). This kind of obturation would create a single interface between the root filling material and root dentin, forming a “monoblock” along the root canal walls, avoiding the creation of gaps between gutta-percha points and sealer (Nagas et al., 2014). In that regard, previous studies have reported that root canal sealers showed higher push-out bond strength values when the obturation was performed in bulk than when it was associated with a master gutta-percha point (Jainaen, Palamara & Messer, 2007; Nagas et al., 2014). Despite these favorable findings, herein, the teeth filled without gutta-percha (AP-B and BC-B groups) showed a similar mechanical behavior compared to those with gutta-percha (AP-GP and BC-GP groups), but still inferior to sound teeth (C− group). Those differences, in groups filled without gutta-percha, could be justified by a thicker layer of sealer, which can shrink during setting and dissolve over time (Jainaen, Palamara & Messer, 2007). These findings contradict the possible advantage of the obturation in bulk once the absence of a main core material may compromise the spreading of the sealer along the root canal walls, facilitating the appearance of bubbles in the obturation mass. When subjected to cyclic loads, the presence of empty spaces in the root canal filling may be detrimental, as they induce stress concentration that could be responsible for the occurrence of fracture.

Regarding the fracture mode, fracture below the cement-enamel junction was the most prevalent pattern. The main reason for this occurrence might be the angle of the applied load, a factor that can influence fracture strength (Krejci et al., 2003). Anterior teeth are loaded in an unfavorable way during chewing and biting since the forces applied to them are oblique (not directed to their long axes) and consequently more harmful (Krejci et al., 2003). In such scenario the stress concentrates at the teeth root juxtaposed to the connection to the supporting tissue (bone in clinic, or the acrylic resin used herein), thus fracture originates on such region, usually discretely below such connection (Wandscher et al., 2014), as shown on Fig. 1. According to our findings, the type of sealer and the filling technique had no impact on the fracture mode.

In vitro studies present inherent limitations; thus, the results of our study should be carefully analyzed. In such sense, we emphasize that the periodontal ligament simulation was not performed in the present study, some authors suggest this reproduction is necessary as it can affect the stress distribution during the fracture strength test (Uzunoglu-Özyürek et al., 2019); however, there others which contradict such recommendation, as its presence may make the fatigue test unstable and thus potentially biasing (Celikten, Uzuntas & Gulsahi, 2015; Dibaji et al., 2017; Missau et al., 2017). That being so, we did not execute such simulation to enhance the quality of load application and stress distribution through the fatigue test.

Conclusion

The use of gutta-percha, as a main core material, associated with the Ah Plus sealer, reestablished the mechanical fatigue behavior of endodontically treated teeth comparable to sound teeth, still consisting on the most promising approach to rehabilitate such scenario. Teeth filled in bulk, had discreetly higher risk of premature failures and inferior fatigue performance. Additionally, while teeth filled with Bio-C Sealer showed a similar mechanical behavior to those filled with AH Plus, they exhibited inferior performance compared to sound teeth. Our work addresses the gap in the literature regarding the fatigue behavior of teeth filled with the new Bio-C Sealer. Future perspectives include exploring improvements in the Bio-C Sealer formulation to enhance its fatigue behavior and conducting in vivo studies to validate our findings under clinical conditions.

Additional Information and Declarations

Competing Interests

Author Contributions

Human Ethics

Data Availability

João Tribst is an Academic Editor for PeerJ.

Isabella Marian Lena conceived and designed the experiments, performed the experiments, analyzed the data, prepared figures and/or tables, authored or reviewed drafts of the article, and approved the final draft.

Luiza Colpo Chiaratti conceived and designed the experiments, performed the experiments, analyzed the data, prepared figures and/or tables, authored or reviewed drafts of the article, and approved the final draft.

Rafaela Oliveira Pilecco conceived and designed the experiments, performed the experiments, analyzed the data, prepared figures and/or tables, authored or reviewed drafts of the article, and approved the final draft.

Renan Vaz Machry conceived and designed the experiments, analyzed the data, prepared figures and/or tables, authored or reviewed drafts of the article, and approved the final draft.

João Paulo Mendes Tribst conceived and designed the experiments, analyzed the data, authored or reviewed drafts of the article, and approved the final draft.

Cornelis Johannes Kleverlaan conceived and designed the experiments, analyzed the data, authored or reviewed drafts of the article, and approved the final draft.

Gabriel Kalil Rocha Pereira conceived and designed the experiments, analyzed the data, authored or reviewed drafts of the article, and approved the final draft.

Renata Dornelles Morgental conceived and designed the experiments, analyzed the data, authored or reviewed drafts of the article, and approved the final draft.

The following information was supplied relating to ethical approvals (i.e., approving body and any reference numbers):

The study was submitted and approved by the Institutional Ethics Committee (CAAE 30345120.7.0000.5346) from the Research Ethics Committee at the Federal University of Santa Maria.

The following information was supplied regarding data availability:

The data is available at OSF and Zenodo:

- Lena, Isabella M. 2024. “Raw Data—The Impact of the Filling Technique with Two Sealers in Bulk or Associated with Gutta-Percha on the Fatigue Behavior and Failure Patterns of Endodontically Treated Teeth.” OSF. September 19. DOI 10.17605/OSF.IO/SW6DF.

- Mendes Tribst, J. P. (2024). Raw data—The impact of the filling technique with two sealers in bulk or associated with gutta-percha on the fatigue behavior and failure patterns of endodontically treated teeth [Data set]. Zenodo. https://doi.org/10.5281/zenodo.13219316.

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
