# Peer review of "The impact of the filling technique with two sealers in bulk or associated with gutta-percha on the fatigue behavior and failure patterns of endodontically treated teeth"

_PeerJ, doi:10.7717/peerj.18221_

## Round 0.1 · original submission · Minor Revisions

Dear authors,
After receiving two reports from external peer reviewers, I have decided that your manuscript would benefit from a minor revision. Please address all the reviewers' comments to avoid further rounds of revision.
We appreciate you submitting your manuscript to PeerJ.

Reviewer 1 ·

Basic reporting

The current study was well conducted and contributes to the existing literature on the topic of filling root canals using the “bulk” technique. Some points that should be addressed, as:

The title of the study is “Sealing techniques and tooth longevity: effects on fatigue behavior and failure patterns.” However, other sealing techniques such as thermoplastic and lateral condensation are not evaluated, and the title suggests a broader range of methods used for obturation.

Figure 1, cited as representative images of failure modes, does not appear in the main document.

Experimental design

Why did you choose to test Bio-C Sealer in the study? Which characteristics of this product justify its use? There are other products classified as bioactive sealers.

Could you provide the details of photopolymerization (J/cm²)? Considering the overall importance of the restoration procedures, this step is highly relevant for the longevity of the restoration.

Validity of the findings

The conclusion could be related to another sealer tested, due to the importance of this material for the study.

Additional comments

The experimental design was well-conducted, and the study report reflects the authors' expertise on this topic. The methods section is detailed, and the results are thoroughly discussed in the context of existing research.
Some points were highlighted to facilitate a better understanding of the topics.

Reviewer 2 ·

Basic reporting

This study evaluated the fatigue behavior of teeth filled with either Bio-C Sealer (BC) or AH Plus (AH), both alone and with gutta-percha. Seventy-two maxillary anterior teeth were tested, with groups including unfilled controls, BC in bulk, BC with gutta-percha, AH in bulk, and AH with gutta-percha. Cyclic fatigue testing showed that sound teeth and those filled with AH Plus and gutta-percha performed best. Teeth filled in bulk had slightly higher failure risks and lower fatigue performance. Using AH Plus with gutta-percha is the most effective for restoring fatigue performance in endodontically treated teeth.

The manuscript is interesting to read. It has good basic reporting.

Minor comment: Please check and correct the font size at lines 85, 220, 260-262, 265, 271, 278-279, 283-284, and 303-304.

Experimental design

The experimental design is comprehensive.

Validity of the findings

MInor comments:

1. The results section is too short, which should be one of the most significant parts of the manuscript. At line 207, the author mentioned that the C- group showed higher FFL and CFF results. Could the author add some explanation here to illustrate what high FFL and CFF mean and why this is significant? I know the author explained some of this in the methods section, but it is not sufficient. The explanation should accompany the results. Similar explanations should be added for the other experimental groups mentioned in the results section.

2. More explanation of the figures should be added in the results section.

3. As the author mentioned in the introduction, "While its physicochemical (Kwak et al., 2023) and biological (Tolosa-Monf‡ et al., 2023) characteristics have been studied, there is still a need for more scientific research on the fatigue behavior of teeth filled with this novel sealer material" (lines 81-83). Additionally, from lines 238-240, the author stated that "These results were also found in a previous study (et al., 2013; Missau et al., 2017), which demonstrated that AH-Plus in association with a gutta-percha point can enhance the fatigue resistance of root-canal treated teeth." Your findings about the AH material are not very novel. BC is the material that hasn’t been studied comprehensively. And author mentioned to compare these two materials at the beginning. So please consider adding more explanation or comparison of these two materials in your results section, making you findings more novel.

4. The conclusion is too short. It should comprehensively conclude your work! Please indicate why your work is specific, what problem it has addressed, and again, what your future perspectives are.

---

## Round 0.2 · Minor Revisions

Dear Authors,

Thank you for submitting the revised version of the manuscript. Although you mentioned that the title was changed as requested by one of the reviewers, it remains unchanged both in the submission system and in the main file. Please correct.

---

## Round 0.3 · accepted · Accept

Dear authros,

We are pleased to inform you that your manuscript, titled "The impact of the filling technique with two sealers in bulk or associated with gutta-percha on the fatigue behavior and failure patterns of endodontically treated teeth" has been accepted for publication in PeerJ following careful review.

The reviewers and editorial team found your work to be a valuable contribution to the field. We appreciate your thorough revisions and efforts in addressing the reviewers' comments.

Congratulations on this achievement!
Best regards,

Reviewer 1 ·

Basic reporting

The text was refined, and the justification for the study was clarified in relation to the selected groups. Additionally, the title was revised.

Experimental design

The issue regarding the details of photopolymerization was clarified.

Validity of the findings

The main question was answered in the conclusion.

Reviewer 2 ·

Basic reporting

The basic reporting is clear and meets all requirements.

Experimental design

The experimental design is comprehensive and meets all requirements.

Validity of the findings

The validity of findings meets all requirements.

Additional comments

The authors have addressed all previous comments well. The revision is good, and I do not have any further comments. It is ready to be accepted.